# Analysis of Adolescent Physical Activity Levels and Their Relationship with Body Image and Nutritional Habits

**DOI:** 10.3390/ijerph19053064

**Published:** 2022-03-05

**Authors:** David Manzano-Sánchez, María Victoria Palop-Montoro, Milagros Arteaga-Checa, Alfonso Valero-Valenzuela

**Affiliations:** 1Faculty of Sport Sciences, University of Murcia, 30100 Murcia, Spain; avalero@um.es; 2Faculty of Health of Sciences, Catholic University of Murcia, 30107 Murcia, Spain; mvpalop@ucam.edu; 3Grupo de Investigación CTS-1018 Physical Activity for Health Promotion, Faculty of Humanities and Education Sciences, University of Jaén, 23071 Jaén, Spain; marteaga@ujaen.es

**Keywords:** physical exercise, physical education, nutrition, body perception, adolescents

## Abstract

The main objective of this research article was to make a cluster analysis in Compulsory Secondary Education students with regard to their physical activity levels, their relationship with nutritional habits and body perception. In this study, a total of 1089 students participated, to whom a battery of tests was given in order to assess three aspects: levels of physical activity, food consumption habits and perception of body image. The main results indicated that the adolescent sample presented high levels of physical activity in comparison with other studies. In addition, a profile analysis was carried out, showing that there were no differences in physical activity, in nutritional habits or in body-image index. Taking into account gender, women who practice light physical activity had better nutritional habits. On the other hand, boys dominated in the group of moderate-to-high physical activity, while the girls were mainly included in the profile of low physical activity. Finally, body-image index was greater in men than women. It was concluded that is necessary to promote the importance of adequate nutritional habits in addition to physical activity, and it is necessary to promote body image, particularly among adolescent girls, given their low values of physical activity and worse body-image perception in relation to boys.

## 1. Introduction

Adolescence is a period of life characterized by numerous biological, psychological and social changes [1]. Many authors divide adolescence into stages: early (between 10 and 13 years), middle (between 14 and 16 years) and late (from 18 to 21 years) [2]. These different stages are susceptible to the acquisition of habits and routines where the choices made relate to the influences of their environment, which can be of significance for their future life, and of great importance in the promotion of healthy lifestyles in this period [3]. These lifestyle habits should include proper nutrition and regular physical activity. Unfortunately, at present there is a loss of optimal eating patterns and a decrease in physical activity in adolescents, who frequently adopt sedentary behavior [4].

In addition, adolescents are influenced by markets and advertising, which help to make their diet unbalanced and excessively caloric, i.e., rich in fats and refined sugars; low in fruits, vegetables, pulses and fish; and in many cases omitting breakfast [4]. Therefore, in 2018 an international study led by the WHO found that one in five adolescents (21%) surveyed were overweight or obese [5] and more recent data on the Spanish population showed that a 34.7% of children and adolescents were overweight, 11.8% obese and 19.5% abnormally obese; the percentage being higher in men than in women [6].

Additionally, the appearance of chronic diseases in adulthood seems to be determined by the existence of modifiable risk factors from an early age [7], and a healthy lifestyle seems to reduce the chances of suffering from cardiometabolic pathologies later in life, suggesting that interventions at these ages is crucial to prevent these diseases [8].

The quantity and quality of physical activity have diminished considerably in today’s society; 25.3% of Spanish adolescents are sedentary [9], 55.2% are inactive [10] and only 34% of those investigated met the minimum international recommendations on the practice of physical activity—the recommendations being 60 min a day, five days a week [11].

On the other hand, body image is defined as the mental and conscious representation that each individual constructs and perceives of their body [12] and is influenced by perceptual, cognitive, behavioral, emotional and cultural elements. It also depends on the self-concept and self-esteem of each person, so it is considered variable during life [13]. Body image in adolescents and non-acceptance of their body representation can lead to greater body dissatisfaction in both genders, due to social and cultural factors [14]. Various authors have observed a higher frequency of dissatisfaction with body image in girls [15,16,17,18,19,20], even at an earlier age, although this increases when they reach adolescence [21].

A systematic review and meta-analysis sought to determine the strength of the associations between physical activity and physical self-concept in children and adolescents. Perceived competence, fitness and physical appearance were strongly associated with physical activity [22]. Sex was a significant moderator for general physical self-concept and age was a significant moderator for perceived appearance and competence. In this way, body image can be one of the most important determinants of eating behaviors and physical activity. Another recent study analyzed the association between body-image perception, nutritional status and dietary and physical activity among adolescents. In their results they found that physical appearance was considered important for almost all adolescents and those who were overweight or obese were unhappy with their weight. Regarding girls, the perception of being overweight or obese was associated with a reduction in the consumption of foods rich in fatty acids and in the performance of moderate-intensity physical activity. In contrast, no association was observed between body image and eating and physical activity among men [23].

Therefore, at this stage of life the acquisition of healthy habits is decisive and they are reflected in the adult stage. Thus, the objectives of this study are to evaluate the levels of physical activity, eating habits and the perception of body image of a sample of adolescents, relating these variables to each other and comparing them by gender.

## 2. Materials and Methods

### 2.1. Sample

The sample of the present study was 1089 adolescent secondary school students: 453 boys (41.6%) and 636 girls (58.4%) (mean age = 14.85; standard deviation = 1.97) belonging to two centers in the Autonomous Community of the Region of Murcia (Spain), The inclusion criteria were: completing all the items in the questionnaire, carrying it out with the researcher and the teacher in class, and meeting the legitimacy criteria of the statistical analysis to detect atypical cases. All participants, as well as their parents or legal guardians in the case of students, were informed in relation to the study, in accordance with the ethical guidelines regarding consent, confidentiality and anonymity of the responses of the ethics committee of research of the University of Murcia (Figure 1).

### 2.2. Design and Instruments

The research was based on a cross-sectional design. The questionnaires were passed to the students in the first minutes of the participating classes. One of the principal investigators was at all times in the class with the teacher, solving any possible doubts that might arise, as well as urging them to answer truthfully, since it would not affect their grades.

The data collection of this work was supported by the favorable report of the Ethics Committee of the University of Murcia (3197/2021). In addition, the parents of the underage students were informed, and these students had to bring a statement of informed consent about the objectives of this study and freedom of participation.

A questionnaire called “Questionnaire on nutrition, body image and physical activity” was prepared, created in Google Forms to be completed in digital format, and divided into five sections: a sociodemographic section, a questionnaire for the analysis of levels of physical activity, another for eating habits, a section in relation to body image and finally, a series of thanks. More specifically, the following questionnaires were used:

*Healthy lifestyle and nutrition questions*: A selection of six questions was made from the “Questionnaires on health-related behaviors and psychological variables” by Tapia-Serrano et al. [24]. These questions were related to the eating habits that influence the health of the adolescents analyzed in the study by Sevil et al. [25]. The questions referred to the frequency of consumption of certain foods, specifically, in relation to the consumption of sugary drinks, juices, fruits and vegetables, French fries and snacks, hamburgers or hot dogs, fish, nuts, sweets and candies. The participants answered the question “How many times a week…?”, indicating seven response options: never, less than once a week, once a week, 2–4 days a week, 5–6 days a week, every day once, every day several times. These values were recoded from 1 to 7 in the statistical program. The scale as a whole had a reliability index of α = 0.71.

*International Physical Activity Questionnaire* (IPAQ): To quantify the physical activity of adolescents, the International Physical Activity Questionnaire (IPAQ) was used. The implementation of the IPAQ began in Geneva in 1998 and has been validated in various studies carried out in different populations and in 12 countries worldwide [26], verifying the reliability and concurrent and criterion validity of the instrument in both its long and short versions. Reliability showed Spearman’s correlation coefficients in the short version, used in the present study; 75% of the observed correlation coefficients were above 0.65 with ranges between 0.88 and 0.32 (r = 0.76; 95% CI: 0.73–0.77). In turn, the criterion validity of the IPAQ data compared with accelerometry showed a moderate-to-high correlation (the short version r = 0.30; 95% CI: 0.23–0.36). The IPAQ researchers developed two versions of the instrument in relation to the number of questions (short or long), the repetition period (“usually in a week” or “last 7 days”) and the method of application (self-administered survey, expansive interview face-to-face or by phone). The questions were coded in such a way that an MET count was performed, classifying physical activity as light (3.3 MET walks × minutes of walking × days per week), moderate (4 MET × minutes × days per week) and vigorous (8 MET × minutes × days per week).

*Body Investment Scale* (BIS) [27]: This instrument is made up of six items that have a 5-field Likert-type response format, where 1 corresponds to “totally disagree” and 5 to “totally agree” related to perception, whether positive or negative of the adolescent’s own body. These six items refer to dimension 1, which includes items related to feelings and attitudes towards body image (for example, “I am satisfied with my appearance”). Items 5, 13 and 17 have the score in the reverse direction. This scale contains 3 other dimensions more related to comfort in contact (e.g., “I enjoy physical contact with others”), with elements about taking care of the body (e.g., “taking care of my body will improve my well-being”), and with items on body protection (e.g., “it makes me feel good to do something dangerous”). The scale as a whole had a reliability index of α = 0.78.

### 2.3. Procedure

The procedure that was developed on the days of data collection in the centers was as follows: Initially, the principal investigator contacted the different centers to show them the objectives of the study and to obtain their approval to administer the questionnaires. Next, while the informed consents of all the students were obtained for participation in the study, a questionnaire was prepared in Google Forms, which was sent to the web of the different centers. This questionnaire was completed by the students in Physical Education classes, always with the same researcher, who was a male Physical Education graduate. The procedure for accessing the questionnaire was explained and some necessary guidelines were given for a better understanding of certain concepts that could lead to error. In addition, they were always asked for commitment, seriousness and to be as sincere as possible. Permission was then given for the students to answer the questionnaire, with the researcher always ready to solve possible doubts. Finally, the students were thanked for collaborating with this study and they were informed of the importance of research in the field of physical activity for their development (Figure 2).

### 2.4. Statistical Analysis

A statistical analysis of the data was carried out using the IBM SPSS 24.0 Statistics Editor. First, the data extracted from Google Forms/Excel was transferred to the SPSS program. The sample was refined by analyzing the missing, maximum and minimum values and the Mahalanobis distance, where a total of 20 participants were eliminated. Subsequently, the different dimensions and variables were elaborated; specifically, the grouping of items to analyze the perception of body image (IBIS), the levels of physical activity (light, moderate and vigorous activity) and the consumption of food (six categories). Next, a normality analysis was performed using the Kolmogorov–Smirnov test for quantitative variables and chi-square for qualitative variables.

After that, the correlation between the variables was checked and a profile analysis was performed using the physical activity of the students as independent variables (low intensity, light intensity and moderate vigorous intensity). To determine the number of profiles, a dendrogram analysis was first performed employing the hierarchical method using Ward’s approach and the most-distant-neighbor method, all obtaining similar results and suggesting the elaboration of two to three sets. Next, a two-stage cluster corroborated a silhouette measure of cohesion and cluster separation, considered good (>0.5) for three sets. Finally, the K-means method was used to make the final three clusters.

Each profile was examined by means of a multivariate analysis of variance (MANOVA) taking into account the differences found in each of the variables under investigation. Additionally, a gender variable was added with two levels: male and female, since it was one of the objectives of our study and can have a significant effect on the measured variables. Finally, the clusters were analyzed according to gender through an analysis of the chi-square value with 2 × 2 contingency tables.

## 3. Results

### 3.1. Descriptive Statistics, Reliability and Correlations

Table 1 presents the mean values, standard deviation, asymmetry and kurtosis values, reliability of the variables and the correlations. The asymmetry and kurtosis values were all <3. The bivariate-correlation test showed positive and significant relationships between most of the variables related to eating habits, and the body-image index was positively related to all of them. The variables related to physical activity were positively related to each other, and the minutes of intense physical activity and walking were negatively related to the consumption of sweets and candies as well as the consumption of chips. 

### 3.2. Cluster Analysis to Obtain Physical Activity Profiles

After removing outliers (Z ± 3 and Mahalanobis distance at *p* < 0.001) we started with the first step using hierarchical-cluster analysis. The dendrogram and the agglomeration coefficients reflected that there were several possible solutions.

The non-hierarchical cluster confirmed the formation of three groups (Figure 3 and Table 2). The first profile was called “moderate and high physical activity” (*n* = 279; 25.6%) with high values in minutes of intense and moderate physical activity per day and intermediate values in minutes of walking; the second profile, called “light physical activity” (*n* = 357; 32.8%), had high values in minutes walked per day and mean values in intense and moderate physical activity; and the third profile, “low level of physical activity”(*n* = 453; 41.6%), had low values in the three variables. On the other hand, Table 2 shows the differences between all the variables that made up the cluster solution. They show a multivariate effect (Box value = 229.310, *F* = 19.029, *p* < 0.001), indicating the violation of the assumption of homogeneity of covariances and suggesting the use of the Pillai trace as a statistical test [28], showing a value of the Pillai trace of 1.229 (*F* = 575.996).

### 3.3. Difference Analysis among Clusters

In order to check whether there were significant differences in the profiles regarding physical activity, a multivariate analysis of variance (MANOVA) was performed. Prior to this analysis, Box and Levene tests were performed to check the previous hypotheses. Box’s test was employed to analyze the homogeneity of covariances, showing that one of the study hypotheses was violated (M de Box = 348.791; *F* = 1.236; *p* = 0.005). The Levene test, which contrasts the equality of variances in the dependent variable in all the groups defined by all the factors, was then carried out, observing that all the *p*-values were greater than 0.05 except for fruit and fish consumption, which means there was homogeneity of variances for almost every one of the variables. Therefore, from the previous tests it was deduced that some of the initial hypotheses were satisfied, but the results cannot be considered completely conclusive.

Once the previous analyses were completed, the multivariate analysis of variance test showed a statistically significant multivariate effect for the gender factor (Pillai’s trace = 0.066; *F* = 7.575; *p* < 0.001) and interactions between the profile and gender factors (Pallai’s trace = 0.039; *F* = 2.130; *p* = 0.002) but not for the three physical activity profile factors (Pillai’s trace = 0.020; *F* = 1.070; *p* = 0.375).

Subsequently, the results were analyzed at the univariate level, showing that for the intersubject gender factor there were significant differences at the level of body-image index (*p* < 0.001), vegetables and greens consumption (*p* = 0.001), and sweets and candies consumption (*p* = 0.001). Regarding the profiles and gender interaction, the univariate analysis showed significant differences at the level of fruit consumption (*p* = 0.017), vegetables and greens consumption (*p* = 0.005), natural juice consumption (*p* = 0.043) and dried fruit consumption (*p* < 0.001) (Table 3).

Since there were significant differences for some variables in the gender factor and interactions between gender and profile factors for many of the variables, it was convenient to analyze the differences between them separately. The post hoc contrast of the comparisons test with Bonferroni’s correction was carried out to determine between which profiles and variables there would be statistically significant differences (Table 4 and Table 5). The analysis of the interactions between gender and profiles revealed there were statistical differences for the light physical activity profile and gender, with higher values for females in vegetables and greens, natural juice and dried fruit consumptions. For the low physical activity profile and gender, significant differences were found for vegetables and greens and sweets and candies, showing higher values for females. For the moderate and high physical activity profile, females showed higher values in sweets and candies consumption (Table 4).

Regarding the comparisons for gender according to the physical activity profile, only females show significant differences, namely that those with light physical activity consume more fruit, vegetables and greens, natural juice and dried fruit than those with low physical activity levels. Furthermore, women with a light physical activity profile also consume more vegetables and greens than those with a moderate and high physical activity profile (Table 5).

### 3.4. Differences in Physical Activity Profiles According to Gender

A Pearson’s chi-square statistic was performed to check if there was any difference in the distribution of the physical activity profiles in terms of gender. Table 6 reports that there were more boys than girls associated with moderate and high physical activity. Most girls were shown to be associated with lighter and lower physical activity levels.

## 4. Discussion

Considering the first of the objectives, high levels of physical activity were observed in our sample of adolescents, unlike other studies that found a decrease in it at this stage of life [29,30,31,32]. Likewise, the HELENA study that measured the level of physical activity and time spent in a sedentary manner in adolescents from nine European countries observed, as in this research, that those who carried out moderate-to-vigorous-intensity activity had high cardiorespiratory fitness and spent less time on sedentary activities, but most adolescents spent most of their time displaying sedentary behavior [33]. It appears that the relative decline of this moderate-to-intense activity affects both sexes from an early age [34], decreases over the years [30,35] and is greater in girls [30,33,34,35,36]. The adequate physical activity level of our study is a good result due to the well-known role of physical activity to prevent premature mortality [37], improve cardiometabolic health [38,39,40,41,42] and aerobic capacity and muscular force [8,43,44].

Regarding the second of the objectives, which sought to relate the levels of physical activity with gender and with the nutritional habits of our sample, the results confirm our hypothesis that male adolescents are the ones who perform more physical activity weekly, though within these, those who carry out intense physical activity showed no differences in eating habits. This is not consistent with most studies, which have shown that adolescents have a deficient consumption of fruit and vegetables [45] or do not eat all main meals [46], except those with high levels of physical activity. Among females, a greater consumption of healthy food was checked in light physical activity (probably to avoid an increase in weight). Unfortunately, the pandemic has forced us to dedicate less time to physical activity outdoors and more time in front of screens (computer, mobile, tablet) favoring the appearance of deleterious effects associated with physical inactivity such as the increase in the consumption of foods with a high caloric intake [47,48,49].

The third of the objectives proposed was to analyze the relationship between levels of physical activity and body image. We must point out that the perception of body image was low in most of the surveyed adolescents. Differences were found in terms of gender. Thus, we found in our sample that men had a more positive perception of their body image, independent of physical activity levels. Studies similar to ours that wanted to verify the association between nutritional status and physical activity with the perception of satisfaction with body image in a sample of adolescents found that being overweight and obese were positively associated with dissatisfaction with body image and the desire to reduce weight in both sexes [50,51].

What is particularly clear in the research on this issue is that the majority of adolescents are dissatisfied with their body image and that this is the group most vulnerable to this perception [12,13,14,15,16,19,20,21,22,23,50,51,52,53,54,55,56,57,58,59,60,61,62,63,64,65]. Physical appearance is considered important by almost all adolescents, as it is a critical stage of life and appearance is one of their greatest concerns [15]. Some authors did not observe an association between body image and eating and physical activity behaviors among male adolescents, but they did observe an association between girls who performed less physical activity and restricted the consumption of certain foods [23]. Thus, body image can be one of the most important determinants of eating and physical activity behavior, leading to the adoption of certain dietary patterns with unhealthy restrictive diets as a solution to the problem [23,52], and/or the practice of inappropriate and extensive exercises [53] or a dependence on exercise [54]. In addition, social pressure provides them with a prototype of a “perfect body” that makes them feel frustrated for not meeting the ideal of beauty, with their self-perception of body image causing psychological alterations such as bodily dissatisfaction or distortion; especially in adolescents with low self-esteem [15].

This is compounded by a recent study showing that body-image disorders are associated with less-moderate and vigorous physical activity and increased screen time [55]. In turn, in this study, subjects with a sedentary lifestyle were more likely to feel dissatisfied than those with active lifestyles, corroborated by other studies where a longer sedentary time was associated with body dissatisfaction in adolescents compared to less than two hours a day of sedentary time [56]. However, contrary to our study, other authors found no association with the level of physical activity [16,23].

Regarding the differences between genders, our research reflects a greater dissatisfaction in girls, in agreement with other studies [56,57,58,59,60]. In contrast, other authors observed a higher prevalence of dissatisfaction with body image among boys than among girls [61], perhaps due to the desire of male adolescents for a more muscular image, since dissatisfaction can be of two types: those who want a greater body image (indicating strength and musculature) and those who want a slimmer image [62]. What most studies agree on, however, is that the pressure exerted by family, peers, television, cinema and fashion is greater in women than in men, already at an early age, although it increases when reaching adolescence [19,23,63,64]. It seems that women are the ones who consume the greatest number of products to improve physical appearance in accordance with cultural ideals, and adolescents may have greater social pressure for a certain body image and perhaps fewer strategies towards criticism and self-acceptance [65].

It should be noted that interventions have been carried out to improve this dissatisfaction through physical activity and nutrition in preadolescents [66] without finding optimal results. There have, however, been attempts to analyze the relationship between the motivations for choosing food and physical activity in the perception of body image among adolescents, finding a preventive effect of the choice of food with respect to corporal satisfaction [67].

As future perspectives, we suggest carrying out studies similar to the present investigation in order to be able to replicate its results and check if they are verified in similar samples (other socioeconomic contexts, different age groups or geographical scope, among other variables). In addition, interventions are recommended with the aim of improving levels of physical activity and the consumption of healthy food, as well as body image, which takes on significant relevance, especially in adolescence. Maintaining physical activity throughout life is challenging, and the consequences of physical inactivity during life have the danger of being transmitted to later generations, creating an intergenerational cycle of poor physical—and even mental—health. Therefore, innovative approaches and new ideas are required on how to improve physical activity levels in children and adolescents [41].

Regarding limitations, we must point out that this study is based on data collected from a cross-sectional survey and that, when using questionnaires for adolescents to report their physical activity (instead of other procedures such as accelerometry analysis) as well as consumption of certain foods, it is possible that some participants showed a degree of bias in their answers or had a greater self-perception of their physical condition than they had in reality. Likewise, the presence of the researcher in the Physical Education classes could influence students’ answers. The variables of the FITT principle for the prescription of physical activity were considered based on intensity and time, but questions about the frequency and type of physical activity performed by the participants could have been included [68] and sedentary behaviors should have been described in similar detail. Specifying these sedentary behaviors constitutes a fundamental challenge to be able to implement strategies aimed at their reduction [69]. Finally, the sample could have been expanded in terms of the age spectrum.

## 5. Conclusions

In conclusion, the adolescents that make up our study sample comply with high levels of physical activity. In turn, there were no differences in body image or nutritional habits in terms of physical activity levels. However, women with light physical activity were the most concerned about taking care of their nutritional habits. The present results may serve for other research to take into account initiatives to promote adequate nutritional habits and not only physical activity. Girls had a worse body-image perception of themselves, which must be worked together with a general improvement in nutritional habits and the promotion of physical activity.

## Figures and Tables

**Figure 1 ijerph-19-03064-f001:**
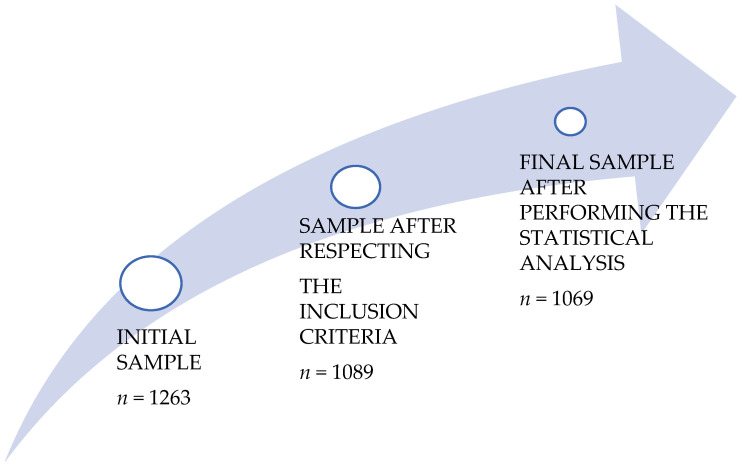
Sample selection process.

**Figure 2 ijerph-19-03064-f002:**
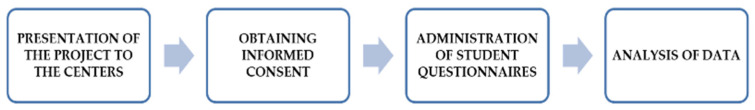
Research process.

**Figure 3 ijerph-19-03064-f003:**
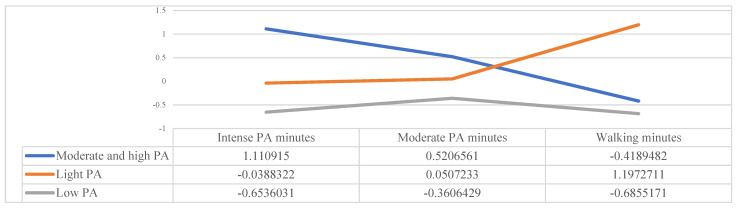
Physical activity levels in students. Note: PA = Physical Activity.

**Table 1 ijerph-19-03064-t001:** Descriptive and correlation values.

		*R*	*M*	*SD*	*A*	*K*	2	3	4	5	6	7	8	9	10	11	12	13
1	Fruit consumption	1–7	4.53	1.86	−0.167	−1.143	0.505 **	−0.048	−0.042	0.351 **	0.310 **	0.006	0.327 **	−0.002	0.111 **	0.106 **	0.105 **	0.080 **
2	Vegetables and greens consumption	1–7	4.57	1.83	−0.200	−1.050	-	−0.019	−0.001	0.230 **	0.299 **	−0.025	0.338 **	−0.052	0.080 **	0.010	0.052	0.072 *
3	Sugary drinks consumption	1–7	2.72	1.80	0.991	−0.035		-	0.421 **	0.208 **	0.109 **	0.485 **	0.081 **	0.312 **	0.090 **	−0.017	0.009	−0.026
4	Sweets and candies consumption	1–7	2.95	1.62	0.809	−0.118			-	0.100 **	0.098 **	0.530 **	0.003	0.316 **	0.097 **	−0.075 *	−0.039	−0.068 *
5	Natural juice consumption	1–7	3.66	1.97	0.264	−1.147				-	−0.401 **	0.209 **	0.305 **	0.151 **	0.156 **	0.088 **	0.077 *	0.072 *
6	Dried fruit consumption	1–7	3.45	1.87	0.434	−0.953					-	0.238 **	0.327 **	0.143 **	0.129 **	0.071 *	0.106 *	0.084 **
7	French fries consumption	1–7	2.198	1.63	0.825	−0.080						-	0.136 **	0.510 **	0.145 **	−0.063 **	−0.050	−0.013
8	Fish consumption	1–7	3.33	1.66	0.342	−0.684							-	0.278 **	0.137 **	0.040	0.091 **	0.043
9	Hamburgers/hot dogs consumption	1–7	2.91	1.46	0.722	0.070								-	0.109 **	−0.013	0.054	−0.014
10	Body-image index	1–5	2.51	0.55	−0.861	0.208									-	0.128 **	0.010	0.016
11	Intense PA minutes per day	-	45.36	41.44	0.737	0.014										-	0.295 **	0.092 **
12	Moderate PA minutes per day	-	34.64	35.00	1.212	1.751											-	0.085 **
13	Walking minutes per day	-	68.12	48.94	0.735	0.110												-

Note: *R* = Range; *M* = Mean; *SD* = Standard Deviation; *A* = Asymmetry; *K* = Kurtossis; PA = Physical Activity; * *p* < 0.05; ** *p* < 0.001.

**Table 2 ijerph-19-03064-t002:** Differences in physical activity minutes from cluster analysis.

	Cluster “Moderate and High Physical Activity”		Cluster “Light Physical Activity”		Cluster “Low Physical Activity”				
	*M*	*SD*	*M*	*SD*	*M*	*SD*	*p*	*eTa*	*F*
Intense PA minutes per day	91.40	32.32	43.75	35.48	18.27	21.31	<0.001	0.495	531.898
Moderate PA minutes per day	52.86	38.98	36.41	33.41	22.01	27.72	<0.001	0.125	77.226
Walking minutes per day	47.63	25.78	126.71	30.47	34.57	22.93	<0.001	0.711	1336.048

Note: *M* = Mean; *SD* = Standard Deviation; *F* = MANOVA effect value; *eTa* = Partial eta squared; PA = Physical Activity; Traza de Pillai = 1.229; (*F* = 575.996); *p <* 0.001.

**Table 3 ijerph-19-03064-t003:** Univariate analysis for gender and factor interaction.

	Gender		Profiles		Gender × Profiles	
	*F*	*p*-Value	*F*	*p*-Value	*F*	*p*-Value
Fruit consumption	1.21	0.27	1.91	0.15	4.07 *	0.02
Vegetables and greens consumption	11.61 **	0.01	0.74	0.48	5.32 **	0.01
Sugary drinks consumption	1.72	0.19	0.59	0.56	0.02	0.98
Sweets and candies consumption	11.31 **	0.01	0.79	0.46	0.66	0.52
Natural juice consumption	1.23	0.27	2.01	0.14	3.16 *	0.04
Dried fruit consumption	2.35	0.13	2.66	0.07	7.72 **	0.01
French fries consumption	0.92	0.38	0.93	0.40	2.19	0.11
Fish consumption	0.45	0.50	2.38	0.09	1.78	0.17
Hamburgers/hot dogs consumption	1.56	0.21	0.43	0.65	0.99	0.37
Body image index	32.596 **	0.01	1.09	0.34	0.31	0.65

Note: *M* = Mean, *SD* = Standard Deviation; *F* = MANOVA effect value; * = *p* < 0.05; ** = *p* < 0.01.

**Table 4 ijerph-19-03064-t004:** Comparisons for gender according to the physical activity profile.

	Male vs. Female × 1	Male vs. Female × 2	Male vs. Female × 3
Fruit consumption	0.215	−0.559 *	−0.046
Vegetables and greens consumption	−0.416 *	−0.868 **	0.099
Sweets and candies consumption	−0.323 *	−0.208	−0.510 **
Natural juice consumption	0.225	−0.504 *	−0.140
Dried fruit consumption	0.251	−0.797 **	0.001

Note: 1 = Low physical activity; 2 = Moderate and high physical activity; 3 = Light physical activity; * *p* < 0.05; ** *p* < 0.01.

**Table 5 ijerph-19-03064-t005:** Multiple comparisons for physical activity profile by gender.

		1 vs. 2	1 vs. 3	3 vs. 2
Fruit consumption	Male	0.210	−0.145	0.355
	Female	−0.564 *	−0.406	−0.158
Vegetables and greens consumption	Male	0.064	−0.321	0.385
	Female	−0.388 *	0.194	−0.582 *
Natural juice consumption	Male	0.214	−0.127	0.342
	Female	−0.515 *	−0.493	−0.022
Dried fruit consumption	Male	0.289	−0.186	0.476
	Female	−0.759 **	−0.437	−0.322

Note: 1 = Low physical activity; 2 = Light physical activity, 3 = Moderate and high physical activity; * *p* < 0.05; ** *p* < 0.01.

**Table 6 ijerph-19-03064-t006:** Differences in physical activity profiles by gender.

		Cluster “Moderate and High Physical Activity”		Cluster “Light Physical Activity”		Cluster “Low Physical Activity”	
		*n*	%	*n*	%	*n*	%
**Gender**	Men	169	60.4	137	38.4	147	27.1
	Women	111	39.6	220	61.6	305	72.9
	*eTa*						0.231
	Chi^2^						57.707 ***

Note: *n* = Size of the sample; *eTa* = Partial eta squared; *** *p* < 0.001.

## Data Availability

https://osf.io/qztx8/?view_only=d7b88aca3e854411b857054258e6f4e1 (accessed on 1 January 2022).

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
