# Peer review of "Analysis of Adolescent Physical Activity Levels and Their Relationship with Body Image and Nutritional Habits"

_ijerph, 2022, doi:10.3390/ijerph19053064_

Round 1

Reviewer 1 Report

This is an interesting research about the Analysis of adolescent physical activity levels and their relationship with body image and nutritional habits.

Here are my comments:

- Lines 10-13 – abstract section - please rewrite the first sentence (the word “their” is used 4 times in 2 and a half lines)

- line 12 – replace “for this” with a more suitable word

- line 30 – “in the” instead of “inthe”

- line 38 – “surveyed adolescents” instead of “adolescents surveyed”

- line 39 - “and 38 this figure was 16.8% inSpanishadolescents” – rewrite, please

- line 39 – who/what is ENPE? Abbreviations should be explained at first appearance in text

- line 40 – “Spanish population between 3 and 24 years of age….” This is a large median of age. Why do you choose this specific study? Your research is about the adolescents’ lifestyle (physical activity, body image, nutritional habits) with possible future direction of study regarding these life choices on future health.

- line 48 – “at these ages is crucial to prevent these diseases” – please try to avoid repeating words

- lines 49-68 – this paragraph should be split and rewrite in a more compact way

- line 69-75 – if these phrases are about reference [22], it should be put at the end of the exposed idea, not at the beginning

- lines 83-90 – maybe you can just present the main objectives. It seems that your hypothesis doubles the information presented in introduction section and decrease the value of your work.

- Overall, I think that introduction section is a litle bit too long and should be rewritten in a more synthesized manner

- lines 93-96 – please rewrite this phrase in order to be understandable and correct incorrect words

- line 94 – please explain “M = 14.65; SD = 1.86”

- line 98 – what do you mean by “meeting the legitimacy criteria of the statistical analysis to detect atypical cases”?

- Design and Instruments section contains many explications that I found not so useful for the readers. You should present the key point of your research, not to describe it gradually. Perhaps it will be useful to create some tables of figures. A flowchart with the included subjects will be helpful for a better understanding.

- I think that the result section is well presented, with exception to minor English mistakes. But I think you should present more clearly the number (percentage) of boys and girls and the age of included subject, or median and IQR.

- Taking into consideration the way you exposed the idea in the discussion section, I think that the first paragraph is useless and should be removed.

- lines 259-278 – I think it will be easier to follow and understand the idea if this paragraph will be split in two.

- line 360 – maybe you can find a better replacement for the phrase “we can conclude by..”

- Teenagers and adolescents are considered the same? Which is the age interval for a boy or a girl to be considered an adolescent? Maybe you can clarify the terms.

Author Response

Dear reviewer.

Thank you so much for your comments. We have attended all your concerns, including the grammar and English mistakes or repeated words.

Here, we attach the specific answer to the other suggestions:

- line  39 – who/what is ENPE? Abbreviations should be explained at first appearance in text

We have explained it in the text the abbreviations.

- line 40 – “Spanish population between 3 and 24 years of age….” This is a large median of age. Why do you choose this specific study? Your research is about the adolescents’ lifestyle (physical activity, body image, nutritional habits) with possible future direction of study regarding these life choices on future health.

This is a large-scale study and the most up-to-date on the percentages of overweight and obesity in Spain. We have collected only the data for the age group that corresponds to adolescence.

- lines 49-68 – this paragraph should be split and rewrite in a more compact way

This paragraph has been shortened.

- lines 83-90 – maybe you can just present the main objectives. It seems that your hypothesis doubles the information presented in introduction section and decrease the value of your work.

We have attend this comment and write only the objectives.

- Overall, I think that introduction section is a litle bit too long and should be rewritten in a more synthesized manner

We have shortened the introduction.

-  line 94 – please explain “M = 14.65; SD = 1.86”

It has been explained in the text

- line 98 – what do you mean by “meeting the legitimacy criteria of the statistical analysis to detect atypical cases”?

This was because the possibility of some students might answer items randomly. It was a statistical test that tried to verify this possibility and exclude this kind of students.

- Design and Instruments section contains many explications that I found not so useful for the readers. You should present the key point of your research, not to describe it gradually. Perhaps it will be useful to create some tables of figures. A flowchart with the included subjects will be helpful for a better understanding.

Two flowcharts have been included with the sample selection process and the procedure followed in the study. In addition, we have suppressed some non-relevant information (figure 1 line 92 and figure 2, line 161).

- I think that the result section is well presented, with exception to minor English mistakes. But I think you should present more clearly the number (percentage) of boys and girls and the age of included subject, or median and IQR.

The number of child has been included

The number of boys and girls has been adjusted. The percentages of boys and girls has been included (the mean and standard deviation of age was already included but we have corrected the figures.

- Taking into consideration the way you exposed the idea in the discussion section, I think that the first paragraph is useless and should be removed.

We have removed this paragraph.

- Teenagers and adolescents are considered the same? Which is the age interval for a boy or a girl to be considered an adolescent? Maybe you can clarify the terms.

In this study, we use the word “adolescents” to the paper and the keyword has been changed.

Reviewer 2 Report

Overall, this study is well done. It very explicit in its details of methods and the script follows the details well. I suspect many readers will not be well versed in MANOVA and how to read/interpret the tables, so I would recommend some explanatory narrative to guide the readers.

Overall, the results are not unexpected. It would be interesting to have some idea about how representative these adolescents are to all of Spain. One analysis that was not done, but would be interesting (and I recommend) is to see if the results are influenced by the gender of the researchers.

The methods tell us that one of the researchers was in the classroom while students filled out the surveys. I assume that each of the 4 researchers (2 males, 2 females) all participated in this. The students' regular teacher was there, of course, but the insertion of another adult might have had some influence on how the students responded. Specifically, were there differences or patterns of response between male students in a classroom depending on whether the attending researcher was male or female, and vice versa for female students.

Adolescents are influenced, directly and indirectly, by the presence of adults. I don't know the age or any other information about each researcher, but I will leave that to the researchers to examine also if they feel there is variation that might be examinable.  But I do think a quick analysis assessing any statistical pattern/relationship between student and researcher gender might not only be informative, and interesting, would also help gauge the findings and interpretations.

Author Response

Dear reviewer,

Thank you so much for your comments. Here, we have attached the specific answer in reference to your suggestions:

- Overall, this study is well done. It very explicit in its details of methods and the script follows the details well. I suspect many readers will not be well versed in MANOVA and how to read/interpret the tables, so I would recommend some explanatory narrative to guide the readers.

Some new text has been added to guide the readers (lines 221-231).

- Overall, the results are not unexpected. It would be interesting to have some idea about how representative these adolescents are to all of Spain. One analysis that was not done, but would be interesting (and I recommend) is to see if the results are influenced by the gender of the researchers.

Taking into account the “Instituto Nacional de Estadística” (INE survey from Spanish Population of 2021), we would need 385 participants with a confidence level of 95% and a 5% error range to extrapolated to the Spanish population.

On the other hand, the researcher was always the same during the completion of the surveys. He was the only one (a man) that was in that process (line 152). This fact could have influenced in the students replies and we have included it in the limitation section.

- The methods tell us that one of the researchers was in the classroom while students filled out the surveys. I assume that each of the 4 researchers (2 males, 2 females) all participated in this. The students' regular teacher was there, of course, but the insertion of another adult might have had some influence on how the students responded. Specifically, were there differences or patterns of response between male students in a classroom depending on whether the attending researcher was male or female, and vice versa for female students.

We have answered this aspect in the previous concern. We are not sure about this problem since he was the only one who said the instruction and answered if there was any doubt.

- Adolescents are influenced, directly and indirectly, by the presence of adults. I don't know the age or any other information about each researcher, but I will leave that to the researchers to examine also if they feel there is variation that might be examinable.  But I do think a quick analysis assessing any statistical pattern/relationship between student and researcher gender might not only be informative, and interesting, would also help gauge the findings and interpretations.

It was always the same person who collected the data, and it was not the purpose of this study to see the influence of the researcher's gender on the result.

Reviewer 3 Report

Dear Authors 

First of all,  English editing is needed to make the research more readable. There are too many repetitions and grammatical errors in the article.

Abstract
Line 10-12; The authors repeated a lot of the word “their” especially in the purpose sentence in this section. The purpose statement should be corrected.

Introduction
-This section has been written fluently by the authors from general to specific, the references used are sufficient and up-to-date.

Line 88, 89-The authors stated the purpose and hypothesis of the research. That's a good thing, but why is there a male-specific hypothesis in their last hypothesis? (I think it should be deleted, the results show this anyway) (If the authors think that it should not be deleted, let them explain with their reasons)

Results
This section is exactly good and understandable.

Materials and Methods and Discussion Section was good agreeing to cross-sectional stıdy, but English editing should make.

Best

Author Response

Dear reviewer.

Thank you so much for your comments. We have taken into consideration all your concerns, including the grammar and English mistakes or repeated words.

Reviewer 4 Report

Thank you for inviting me to review this very interesting manuscript. The study is very well designed, based on a sample of more than 1000 Spanish adolescents, uses multivariate statistics for data analysis and covers three interrelated areas of a healthy lifestyle: physical activity, nutritional habits and body image. All sections of the manuscript are well developed and correctly described. The study is innovative and has clearly stated research objectives. However, I suggest the following recommendations to further improve the quality of the manuscript:

Introduction:

  • Line 27: Please briefly describe the stages of adolescence mentioned.
  • Line 37 mentions the 2012 HBSC study - it would be better to replace this with the recently published findings of the HBSC study.
  • The introduction would be more meaningful if the paragraphs were separated by topic (e.g. body image in lines 60-61 should be moved to the new paragraph).

Methods:

  • Section 2. Materials and Methods (line 91): please indicate the research design of your study.
  • 2.1 Procedure: it is important to clearly state that the data collection was conducted on site with an online survey.
  • Line 176: 'statistical treatment' - it would be better to use another word, e.g. 'statistical analysis'.

Results:

  • Although the methodology is justified and convincing, reconsider the data analysis. A similar cluster analysis could be done for the nutritional measures as presented for physical activity. This would allow for a comparable analysis. It is not clear why the nutritional items in the data analysis were used directly from the measurement and not in a generalized form of adolescent nutritional habits (e.g. cluster, index, component, factor) compared to the other two areas studied (physical activity, body image);
  • To examine the difference between female and male participants, please consider all three perspectives (physical activity, nutrition habits and body image) simultaneously. A suitable approach for this is discriminant analysis. This method includes more dependent variables and shows how these variables define or separate at least two different groups (e.g. women, men). I see your research question as strongly related to this method because of the three equally important perspectives (physical activity, nutrition and body image). You could use a discriminant analysis to find out which components contribute significantly to the differences between the genders. Table 5 ( line 253) shows only the difference in physical activity. The other two components analysed are missing, although they are mentioned in the research hypothesis ( lines 88-90).

Discussion:

  • Consider excluding theoretical parts (e.g. line 268: 'lack of physical activity is a global public health problem' or lines 268-278 about the impact of physical activity on health). These parts fit better in the introduction section.
  • Regarding limitations, the FITT principle of physical activity should be mentioned. In this study, only intensity and waking time (type) were used to define groups of adolescents according to their physical activity. The other characteristics of physical activity do not seem to have been considered in the data collection/analysis. This should be mentioned and justified in the discussion.

Technical comment:

  • Please check the whole article carefully for technical errors - e.g. some words are written together (e.g. line 39: "inSpanishadolescents"; line 104: "wasbased").

Round 2

Reviewer 2 Report

This is a good paper, project well done. The changes you made will help readers follow and digest the results.

Author Response

Thank you.

Reviewer 4 Report

Dear Authors,

thank you very much for your valuable work and careful consideration of the suggested improvements. The Introduction section is now much clearer and warranted. Thank you very much for the explanation of the statistical analysis and results presentation. The manuscript quality was improved. However, I would like to suggest two additional minor revisions:

  • Figure 1: the size of the cycles is increasing meanwhile the size of the sample decreases. Probably, it should be the opposite.
  • I am still not sure about the Table 5: In my opinion the gender analysis has not so much sense at the end of the results section. At these point the reader is expecting to consider all three previously mention perspectives – physical activity, nutrition and body image. I don’t see much value to expose only the gender differences at this point. Please, consider to use for comparative analysis any other statistical analysis if discriminat statistics do not fit well to your research objective. The second option is to include findings on gender differences in the interpretation of the Table 2.

Thank you very much for your efforts!

Author Response

Dear reviewer,

Thanks to your suggestions we have done a new Manova analysis including gender next to the physical activity profiles factor from the beginning. This has help us to identify if there was any change in the variables due to the gender apart from physical activity and also, if there was interaction between gender and physical activity profiles. We have built two new tables and some text describing these new outcomes. These new analysis has reported new results and we have taken into account all of them for rewritting the discussion, conclusion and abstract sections. We expect all of this can give an appropiate answer to your requirements.